

# Atmospheric forcing of sea ice anomalies in the Ross Sea Polynya region

Ethan R. Dale[1, 2], Adrian J. McDonald[1], Jack H.J. Coggins[1], and Wolfgang Rack[2]

[1]Department of Physics and Astronomy, University of Canterbury, Christchurch, New Zealand.
[2]Gateway Antarctica, University of Canterbury, Christchurch, New Zealand.

*Correspondence to:* Ethan Dale (ethan.dale@pg.canterbury.ac.nz)

**Abstract.** Despite warming trends in global temperatures, sea ice extent in the Southern Hemisphere has shown an increasing trend over recent decades. Wind-driven sea ice export from coastal polynyas is an important source of sea ice production. Areas of major polynyas in the Ross Sea, the region with largest increase in sea ice extent, have been suggested to produce a vast amount of the sea ice in the region. We investigate the impacts of strong wind events on the Ross Sea Polynyas and its sea ice
concentration and possible consequences on sea ice production.

We utilise Bootstrap sea ice concentration (SIC) measurements derived from satellite based, Special Sensor Microwave Imager (SSM/I) brightness temperatures. We compared these with surface winds and temperatures from automatic weather stations (AWS) of the University of Wisconsin-Madison Antarctic Meteorology Program. Our analysis focusses on the austral winter period defined as 1st April to 1st November in this study. Daily data were used to classified into characteristic regimes
based on the percentiles of wind speed. For each regime, a composite of SIC anomaly was formed for the Ross Sea region. We found that persistent weak winds near the edge of the Ross Ice Shelf are generally associated with positive SIC anomalies in the Ross Sea Polynya (RSP). Conversely we found negative SIC anomalies in this area during persistent strong winds. By analysing sea ice motion vectors derived from SSM/I and SSMIS brightness temperatures, we find significant sea ice motion anomalies throughout the Ross Sea during strong wind events. These anomalies persist for several days after the strong wind
event.

Strong, negative correlations are found between SIC and AWS wind speed within the RSP indicating that strong winds cause significant advection of sea ice in the region. We were able to recreate these correlations using co-located ERA-Interim wind speeds. However when only days of a certain percentile based wind speed classification were used, the cross correlation functions produced by ERA-Interim wind speeds differed significantly from those produced using AWS wind speeds. The rapid
decrease in SIC during a strong wind event is followed by a more gradual recovery in SIC. This increase occurs on a more gradual time scale than the average persistence of a strong wind event and the resulting sea ice motion anomalies, highlighting the production of new sea ice through thermodynamic processes. In the vicinity of Ross Island, ERA-Interim underestimates wind speeds by a factor of 1.7, which results in a significant misrepresentation of the impact of winds on polynya processes.



## 1 Introduction

Throughout the satellite observation era the total winter sea ice cover in the Southern Ocean has followed a well established increasing trend, a process which is mainly driven by significant sea ice growth in the Ross Sea (Comiso and Nishio, 2008; Turner et al., 2009; Holland, 2014; Turner et al., 2015). However, there is still uncertainty about the mechanisms which have driven this change. The central aim of this work is to study how the variability of strong southerly winds over the western Ross Ice Shelf impacts sea ice production and transport in the region near the Ross Sea Polynya (RSP). A polynya is an area of open water or decreased sea ice concentration (SIC) surrounded by either concentrated sea ice or land ice. Due to the potential for increased ocean to atmosphere heat transfer within these regions, polynyas are areas of high sea ice production (Tamura et al., 2008). The Ross Sea Polynya is a large polynya that regularly forms near the northwestern edge of the Ross Ice Shelf as a result of persistent offshore winds.

This work builds on previous studies, such as by Kwok et al. (2007), Reddy et al. (2007), Comiso et al. (2011), Drucker et al. (2011), Holland and Kwok (2012) and Turner et al. (2015), who have shown that the Ross Sea Polynya and the related atmospheric forcing plays an important role in sea ice production. For example, Drucker et al. (2011) estimates that 20-50% of the sea ice production in the Ross Sea occurs in the RSP and that the increase in sea ice extent in the Ross Sea region is related to the changes in the wind forcing.

Previous studies have been constrained by a lack of detailed measurements. The weather patterns over the Ross Sea contain many small scale features that are governed by the topography of the area. Thus, current models such as the Antarctic Mesoscale Prediction System (AMPS - Powers et al. 2012) are unable to resolve many of these features (Coggins et al., 2013; Jolly et al., 2015 (in review). This means that understanding of the direct influence of strong winds on the formation of the RSP, a region of major sea ice production has been limited. The analysis of the seasonal patterns of Antarctic sea ice growth and decline and its inter-annual variability is complicated by the fact that they depend on a number of atmospheric and oceanic forcings which occur at a wide range of time scales. In particular, the extent of sea ice is influenced by both atmospheric and oceanic factors, including the strength of the near-surface winds, air temperature, ocean currents, ocean temperature and salinity of the ocean (Bintanja et al., 2013; Holland and Kwok, 2012; Holland, 2014; Turner et al., 2015).

Coastal polynyas, such as the RSP, are triggered by sea ice export from the coast with sea ice drift being controlled by both oceanographic and atmospheric forcings. Ice in free drift will have a velocity equal to that of the local ocean current plus some component due to the effect of wind stress. In the Southern Hemisphere this wind component will fall to the left of the wind vector and has a magnitude up to about 2% of the local wind speed (Brümmer et al., 2008). In consolidated ice internal stresses will oppose the geostrophic wind and therefore decrease ice drift velocity (Brümmer and Hoeber, 1999).

Sea ice concentration (SIC) is defined as the percentage of the ocean covered by sea ice for a given area. In this study, sea ice extent (SIE) is defined as the integral of all pixels where SIC is greater than 15%. While Arctic sea ice extent and areal coverage has displayed a clear decreasing trend over the satellite period, observations in the Antarctic show the opposite tendency (Turner et al., 2007). Changes in the area of the Antarctic sea ice have been less dynamic with a small, but statistically



significant overall increase in both ice extent and area throughout the year, although this general increase masks larger opposing regional trends (Comiso and Nishio, 2008; Maksym et al., 2012).

The primary synoptic-scale atmospheric variations affecting sea ice include the overall magnitude of the geostrophic wind (Sen Gupta and England, 2006), the localised zonal and meridional wind anomalies (Stammerjohn et al., 2008; Kwok and
Comiso, 2002; Sen Gupta and England, 2006; Turner et al., 2009; Holland and Kwok, 2012), surface air temperature anomalies (Sen Gupta and England, 2006; Kwok and Comiso, 2002) and variations in energy fluxes between the atmosphere-ocean-sea ice systems. Geostrophic winds are also central to describing the variations in localised Ekman transport patterns within the ocean. Specifically, Stammerjohn et al. (2008) identify that enhanced Westerlies throughout the 1990s in the western Ross Sea caused a more persistent northward Ekman sea ice drift, which affected the seasonal ice extent of the region by causing earlier
ice advance and later ice retreat.

This work therefore has a wider relevance given that atmospheric circulation changes in the Ross Sea may explain a significant portion of the climate variation in the region and particularly increases in sea ice extent and the northward drift of sea ice (Holland and Kwok, 2012; Nicolas and Bromwich, 2014). Holland and Kwok (2012) used sea ice motion data and reanalysis wind fields to show that wind-driven changes in ice advection are the dominant driver of SIC trends around much of West
Antarctica. By contrast, wind-driven thermodynamic changes play a large role in coastal regions of the Atlantic sector (Kong Håkon VII Hav Sea) where autumn SIE trends oppose the near-surface wind variations.

In this study, we investigate winter in-situ measurements from weather stations directly upwind of the Ross Sea Polynya in relation to satellite measurements of sea ice cover in order to better understand the time scales over which surface wind impacts on sea ice drift and sea ice production. We do this in comparison with winds from a low resolution reanalysis model (ERA–
Interim (Dee et al., 2011))in order to find out to what extent simulated wind fields in the region can reproduce the statistical relationship and dependence between weather and sea ice anomalies.

## 2   Data and Methods

We utilise both Basic Bootstrap Algorithm and AMSR-E Bootstrap Algorithm sea ice concentration (SIC) data (henceforth Bootstrap collectively) provided by the National Snow and Ice Data Center (NSIDC) (Comiso, 2000; Maslanik and Stroeve,
2004). Bootstrap SIC is available on a daily basis sampled onto a 25 km by 25 km grid since 1987 and every other day prior to this date (starting in 1977). Daily averages of SIC swath data is provided by NSIDC, these were assumed to be centred around midday with measurement spread 12 hours before and after the reported time. We obtain 10 m wind speed and 2 m temperature data measured at the Laurie II automatic weather station (AWS), located at 77.52° S 170.81° E provided by the University of Wisconsin-Madison Automatic Weather Station Program detailed in Lazzara et al. (2012). The Laurie II station
has been providing data at 10 minute temporal resolution since February 2000. In this study we focus on output from the Laurie II station, because of its long continuous record and proximity to the RSP. The analysis presented has also been completed for a number of other stations, including Vito, Emilia and Ferrell stations, and these results remain rather similar.



The standard deviation derived as the daily variation around the inter-annual mean of Bootstrap SIC from the 20th April until the 1st of November for years 1979 until 2014 is presented in Fig. 1. Coastal pixels often display a large variability and a prominent area of variability is located between longitude 170° E and 180° extending several hundred kilometres offshore. The orientation of this area corresponds well with both the dominant wind directions observed at the Laurie II AWS site and

the location of the RSP as identified in Nakata et al. (2015) amongst others. Large deviations from the mean are due to the high variability in SIC within the polynya displayed in Fig. 2.

For our analysis we define a region comprised of pixels adjacent to the Bootstrap land mask and 10° wide in longitude, centred on the Laurie II AWS site, identified in red in Fig 1. This gives a 25 km by 250 km region close to the Ross Ice Shelf and Ross Island co-located within the area where the Ross Sea Polynya can be observed. The wind roses in Fig. 1 display the

distribution of wind vectors observed at various AWS sites (Margret, Vito and Laurie II). Inspection of the wind rose closest to Ross Island (Laurie II) shows that the wind in this region is dominated by southerly flows with a third of the observations linked to high ($> 7.5 \ ms^{-1}$) wind speeds. These strong southerly flows are also an important feature of the wind distribution at the other AWS sites displayed in Fig. 1, but are not observed as frequently. However, as identified previously the results presented do not change appreciably if data from other AWS sites in the northwestern corner of the Ross Ice Shelf is utilized.

We derive sea ice motion vectors from the National Snow and Ice Data Centre's 12.5 km resolution polar stereographic gridded brightness temperatures, retrieved from the Special Sensor Microwave/Imager (SSM/I) and Special Sensor Microwave Imager/Sounder (SSMIS) instruments (Maslanik and Stroeve, 2004). Daily averages are available from July 1987 to December 2015. We employ the vertical and horizontal polarisations of the 85.5 GHz channel from 1987 to 2009 and the 91.7 GHz channel from 2010 onwards.

Following multiple authors (Emery et al., 1997; Heil et al., 2006; Holland and Kwok, 2012), we estimate ice motion via a maximum cross-correlation method. We track $9 \times 9$ grids of stereographic cells ($112.5 \times 112.5$ km) over a radius of 8 grid cells (100 km). The search radius provides an upper limit of ice velocity of approximately 1.2 ms$^{-1}$, defining a physically plausible range (Heil et al., 2006).

The cross-correlation method proceeds by comparing consecutive daily averages of brightness temperature data. For each

day, brightness temperatures in a particular grid are correlated against those in the surrounding grids on the subsequent day. The grid with the highest correlation is then designated as the grid to which the motion has occurred. A threshold of 0.7 is placed on the correlations to limit erroneous designations. Our method uses the maximum correlation from both the horizontal and vertical polarisations. From the consequent estimate of displacement, an estimate of ice velocity in the intervening period can be attained. We subsequently smooth the gridded velocities to a 25 km stereographic grid.

It is important to note that there are some motions which will not be apparent using this method. For instance, fast small-scale motions are below the resolution of the grid and will not be captured. Small scale rotations and divergence of the ice pack will not be observed as the cross-correlation method does not account for these types of drift. Further, motion in coastal areas is likely to be inaccurate due to the difficulties in applying the method to incomplete grids.



Brightness temperature derived motions are considered inaccurate outside of the winter season due to surface melt and high levels of atmospheric water (Emery et al., 1997; Holland and Kwok, 2012). Hence, we restrict analysis of sea ice motions to the Antarctic cold months of April to November.

We compare winds observed at the AWS sites with the ERA-interim meteorological reanalysis model (Dee et al., 2011).

The model output is available on a $0.75° \times 0.75°$ grid at a 6 hour temporal resolution running from late 1979 until present. ERA-interim does not assimilate wind speed measurements over land (including ice shelves) and is therefore independent from AWS measurements. For comparison of AWS and ERA-interim wind data virtual AWS stations were created by interpolating the wind speed from the ERA-Interim grid to the location of the AWS sites using a bilinear interpolation.

## 3   Results

Figure 2a shows the mean SIC within the coastal area identified in Figure 1 averaged over the period 1988 to 2014. Throughout the winter period, defined between the first of April to the first of November in this study, the total SIC within this area is relatively constant. Outside this period we observe a gradual decrease from November in SIC until a minimum is reached in mid-February followed by a more rapid increase in SIC as the sun sets in early March. For the remainder of this analysis we will only consider the period from the first of April to the first of November to remove the effects of summer melt and to avoid

periods with low SIC and large gradients in SIC. Figure 2b shows the daily SIC for the same area derived from data in 2013 to show the high degree of variability in the SIC in this region, these results resemble previous analysis reported in Bromwich et al. (1998). The large day to day variability between April and November (Fig 2b) are likely to be associated with polynya processes. The sawtooth features observed in this specific year, for example around 1st May, are common features and illustrate that decreases in SIC generally occur more rapidly than the following increases in SIC, a point that will be supported by later

analysis.

The wind speeds measured at four AWS, Laurie II, Ferrell, Emilia, and Vito were compared to the virtual sites interpolated from the ERA-Interim model grid. Scalar wind speeds at Laurie II, Ferrell and Emilia correlated well with $R^2 > 0.75$ while Vito was found to have a weaker correlation of $R^2 = 0.55$. However when linear least squares fits between the model and the Laurie and Ferrell AWS winds were applied, slopes of 1.70 and 1.52, respectively, were found, indicating that at these sites

ERA–Interim generates significantly weaker wind speeds than measured by the AWS. Scale factors measured at Emilia and Vito were 1.06 and 0.96 respectively, indicating a better agreement between the AWS and ERA-Interim wind speeds at these sites. Laurie and Ferrell lie 36 and 57 km from Cape Crozier (at the eastern end of Ross Island), while Emilia and Vito lie 140 and 223 km east of Ross Island. Thus, the differences observed are likely linked to the local topography that is not well represented in the ERA-Interim reanalysis. Recent work by Jolly et al. (2015 (in review) comparing AWS observations with

the Antarctic Mesoscale Prediction system output (a much higher resolution atmospheric model) also identifies that the effect of topography in the region can not be reproduced by the model (Powers et al., 2012) . It should also be noted that ERA-Interim does not assimilate wind speed measurements over land and so the two datasets are independent Dee et al. (2011).



Cross correlations functions (CCF) between the time series of SIC within the region in Fig 1 and both wind speeds and temperatures measured at the Laurie II AWS site were calculated. The wind stress on sea ice depends on the square of the wind speed. This does not, however result in a linear relationship between SIC and the square of the wind speed because wind stress only defines the force causing the advection. For this reason wind speed was correlated with SIC rather than the square

of wind speed. For comparison with daily Bootstrap SIC data 24 hour running means of the 10 minute AWS measurements were used. Although SIC was only available on a 24 hour resolution, wind data was available at a 10 minute resolution. By varying the time lag between these two time series and calculating the Pearson correlation coefficient for each lag, CCFs were able to be calculated at a 10 minute time resolution. The CCF between the time series of SIC and both scalar wind speeds and temperatures measured at the Laurie II AWS site are shown in figure 3. In figure3 (a) we find a strong negative correlation

between SIC and wind speed with the maximum magnitude correlation occurring after a 12 hour time lag. The minimum is preceded by a rapid decrease and followed by a more gradual increase in correlation with respective e-folding times of -48 hours and 100 hours. This indicates that during high wind events the SIC in the RSP-area is generally low. The difference in the decrease and increase e-folding times suggests that the two changes are controlled by different processes. For example we expect that the decrease in SIC is dominated by more rapid dynamic processes while the increase in SIC during sea ice

formation is dominated by slower thermodynamic processes. This interpretation is supported by the recent analysis detailed in Nakata et al. (2015) which used a simplified model to understand polynya changes. Laurie II is located about 50 km south of the region used for the SIC analysis. A change of predominantly southerly winds (Parish and Bromwich, 2007) is first observed upwind at Laurie II before the signal propagates to the RSP downwind. Wind blowing at $5\,\mathrm{ms}^{-1}$ wind would take about 3 hours to travel the distance so the 12 hour delay observed cannot be entirely explained by this process. Another contributing factor

is that the region used for calculating the SIC is not directly adjacent to the northern edge of the RIS and an area of sea ice will exist south of the studied polynya area. If a northward advection of sea ice occurs the ice advected from the region will be replaced by ice in the unobserved region between the Ross Ice Shelf and the region specified in Fig. 1. This could allow advection of sea ice to occur for a short period of time without a decrease in SIC occurring, causing the minimum in sea ice to occur several hours after a significant increase in wind speed. In addition, as a strong wind event progresses the sea ice will

ridge and raft upon itself, causing the surface roughness of the ice to increase making the ice more susceptible to the wind stress (Mårtensson et al., 2012). As the SIC decreases, internal stresses prohibiting motion will decrease allowing advection to become more significant. These effects will cause the minimum SIC to occur slightly after the time of maximum wind speed and are likely causes for the location of the minimum correlation between SIC and wind occurring at approximately 12 hours delay, though the collection of swath data also introduces some uncertainty in this exact timing. Winds in the area autocorrelate

with an e-folding time of 36 hours (Fig 3a), which explains the significant correlations at negative delay, as the e-folding time for the decrease of the wind, the SIC cross-correlation function is observed to minimise at 48 hours and the extrema occurs at 12 hours delay.

The temperature versus SIC CCF also shows a significant negative correlation with the minimum occurring at 0 hours delay (Fig 3b). The e-folding times for this curve are -120 hours and 170 hours for the decreasing and increasing periods,

respectively. These values are both similar to the autocorrelation e-folding time for temperatures measured at Laurie II of 150





hrs. As a strong wind event progresses and SIC decreases, the heat flux from the ocean to the atmosphere will increase causing production of new sea ice through freezing. The rate of freezing of open water will depend on the temperature differential between the ocean and the atmosphere. Thus, potentially explaining the significant correlations observed. However, significant positive correlations can also be found between the temperature and wind speeds measured at Laurie II station (Fig 3b), which

we interpret as at least partially the cause of the correlation between temperature and SIC. In addition, work detailed in Coggins et al. (2014) also suggests that strong wind events, particularly Ross Ice Shelf Air Stream (RAS) events (Parish et al., 2006), are related to warm surface temperature anomalies. However, given that the correlation between temperature and wind speed at Laurie II is 0.5 at zero hours and the correlation between temperature and SIC is -0.65 there must be some other causal link between temperature and SIC. We suggest that this is likely due to a more gradual rate of freezing of open water during

periods of higher air temperatures (effectively a smaller temperature differential between ocean and air temperatures impacting the sensible heat flux).

CCFs were also produced using the ERA–Interim virtual AWS stations in the same manner although at a 6 hour temporal resolution dictated by the temporal resolution of the ERA-Interim output. We found that both the wind speed, SIC and wind speed autocorrelation were very similar to that found using the AWS wind speeds (Fig 2a). This is not surprising as the

ERA-Interim wind speeds correlate well with that of AWS at Laurie. The temperature versus SIC CCF, temperature versus wind speed CCF and the temperature autocorrelation from the ERA-Interim output have similar forms to the relationships derived using AWS data, but are generally smoother and the magnitude of the largest correlations are generally smaller. The temperature autocorrelation curve shows that ERA-interim predicts more persistent temperatures than the Laurie II AWS measures. This likely indicates that high frequency temperature fluctuations are not accurately modelled within ERA-interim. The

ERA-Interim wind speed versus temperature CCF shows weaker correlation for delays less than 24 hours with the difference becoming negligible around -72 hours. This probably indicates that the warming of the air due to mixing, suggested in Coggins et al. (2013) and other studies, caused by strong winds is underestimated in ERA-interim. These effects likely cause the small differences between the ERA-interim temperature and SIC correlations and that of Laurie II AWS.

Cross correlation curves for SIC and wind speed measured at Laurie II AWS site and wind speed hindcast by the ERA-

Interim virtual station for low, medium and high wind classes defined using thresholds calculated from the AWS data are now examined in figure 4(a-c). We will begin by discussing the AWS results and then follow with a comparison of these results with those found using the ERA-interim virtual station. The daily mean wind speeds measured at Laurie II were categorised into low, medium and high wind events based on 33rd and 66th percentile AWS wind speeds measured at Laurie II (3.5 ms$^{-1}$ and 7.6 ms$^{-1}$ respectively). Wind direction was not considered in this classification because of the predominance of southerly

flows at that site. CCF's for SIC with wind speed were calculated only for periods when the mean wind speed during a -12hrs to +12hrs period was within one of the three categories (Fig 4). Autocorrelation curves for the wind in each of these three cases were also calculated to allow the persistence of each of the three strengths of wind speed events to be identified. The AWS medium wind case autocorrelation curve (Fig 4b) shows considerably lower persistence than either of the extreme cases, indicating that this is a transition state that occurs frequently for short periods. The high wind case (Fig. 4c) shows a CCF

very similar to that of all cases (Fig 3b) excluding the period from -36 hours to +12 hours delay. If we assume the relationship





between wind speed and SIC is linear, splitting the data in this manner will result in lower correlations in each individual state when compared to the total. Effectively, a smaller range of wind speeds is sampled, while the variability of the SIC remains constant therefore increasing the uncertainty and reducing the correlation. However, the force exerted on sea ice is proportional to the wind stress, which is proportional to the square of the wind speed, which means that the winds within the high wind

case will have a greater impact on sea ice motion than those of the lower cases. The low and medium cases both show negative correlations between SIC and wind speed, except for a short period, spanning -30 hours until 6 hours where weak positive correlations are observed. The medium case has a correlation extrema of -0.5 at 30 hours while the extrema for the low case is -0.4 at 50 hours between AWS and SIC data. Both these values are significantly weaker and later than that of both the high and total cases. This could be because weaker wind speeds will not have as significant effect on sea ice causing any advection of

sea ice and subsequent sea ice break up to occur much less rapidly than the high wind case. Another possible cause is that the higher wind speeds within each class are more likely to increase to stronger cases after the classified period, causing a decrease in SIC at a large delay. Due to the autocorrelation for both medium and low cases being very low at their respective times of extrema the former explanation seems unlikely as there would be little coherence between wind speed at 0 hours and wind at the extrema.

The categorised autocorrelation curves found using the ERA–interim virtual station are not as perfectly symmetrical as those for the AWS data. This is due to errors that occur at the beginnings and ends of the broken time series obtained because of the wind speed classifications used. These errors are also present in the AWS autocorrelation curves, but are of greater significance when using the low temporal resolution ERA-interim data. The medium ERA-interim wind speed autocorrelation curve shows persistence similar to that of the low and high cases differing from that of AWS which showed a much shorter persistence in

the medium case. The low case wind, wind speed versus SIC CCF derived from ERA-interim output is very similar to that found using AWS. In contrast the ERA-interim CCF's in the medium and high wind speed cases differ significantly from that of AWS. In particular, the ERA-interim medium case displays stronger negative correlations than that of the AWS between -24 and 24 hours after which the two are very similar. The high wind regime for ERA-Interim displays significantly weaker correlations between -12 and 72 hours than the corresponding pattern derived using AWS data. The latter point likely reflects

the fact that the ERA-Interim data is generally poor at representing the strength of the wind in the strong wind speed periods for this region, this being supported by the large gradient derived (1.70) when applying a linear least squares regression to the ERA-Interim and AWS wind speeds.

To gain a greater understanding of the influence of winds on SIC in the region, we now consider the spatial structure of the SIC anomalies for the high and low wind classes previously identified for different periods relative to the onset of those classes.

For each of the low and high wind classes, the 1st April to 1st November mean Bootstrap SIC anomaly for each pixel in the Ross Sea region was calculated over the 2001-2014 period (Fig 5). Composites for several days of delay before and after the wind event onset were then derived to highlight how the sea ice anomaly varies spatially prior to and following these wind classes. Histograms are also shown to indicate how the distribution of each wind class changes throughout the period examined. We find significant, positive anomalies within the Ross Sea Polynya during low winds and negative anomalies during high winds

in general (Fig. 5). No significant anomalies were found during the medium wind cases and these are therefore not displayed.





All significant anomalies found occur within known polynyas, this is likely because sea ice is generally thinner and has a lower concentration within polynyas. Significant anomalies are observed 2 days before the measured wind and remain until 5 days after, an imbalance in the proportions of the three wind classes (indicated by the inset histograms) also remains for a similar period. The SIC versus wind speed CCF displayed increase and decrease e-folding times of -48 and 100 hours (Fig. 3). Thus

during a high wind event the period where SIC is decreasing seems to be significantly impacted by advection.

To further our understanding of the role of sea ice advection, sea ice motion vectors were derived from 85 and 91 GHz band brightness temperatures. The mean sea ice motion vectors for the 1st April to 1st November period from 2001 until 2014 are shown in Figure 6. Figure 6 shows northward flow throughout the Ross Sea with an easterly component occurring to the east of Cape Adare. This highlights the net export of sea ice from the north facing coasts of the Ross Sea throughout this period

(Comiso et al., 2011). Composites of sea ice motion anomalies related to high and low wind states at delays varying from -2 days to 3 days from the wind event are displayed in Fig. 7. During periods of low wind speed at Laurie II, anticyclonic anomalies occur throughout the Ross Sea (Fig. 7(a-f)). Conversely cyclonic anomalies are found during periods of high winds at Laurie II (Fig. 7(g-l)). These anomalies are largest at the time that the wind state is identified, but persist for 24 hours after the wind event with weak anomalies being found in both low and high states 48 hours after the wind event. It is also noticeable

that no coherent pattern in the sea ice anomalies associated with the medium wind state are observed (not shown), highlighting the critical influence of atmospheric near-surface winds on sea ice motion in the region.

## 4 Discussion

ERA-Interim was able to generate wind speeds that correlated well with that of several AWS sites, indicating that the relative wind speeds within ERA-Interim were consistent with the AWS measurements. However, at AWS sites near significant

topography ERA-Interim was found to predict wind speeds significantly weaker than the winds measured by the AWS with scale factors of 1.70 and 1.52 (implying that the ERA-Interim values are this factor smaller than the AWS measurement) at the Laurie II and Ferrell sites, respectively. This is likely because ERA-Interim is unable to accurately model the mesoscale barrier affect of Ross Island and the resulting flow convergence. A hypothesis that is supported by recent comparisons between AWS data and mesoscale model output in the region (Jolly et al., 2015 (in review). The wind speed versus SIC CCF produced using

the virtual Laurie station was very similar to that found using the Laurie AWS data. However, when the data was separated into low, medium and high wind regimes, based on a categorisation derived from the AWS data, significant differences were found between the ERA-Interim and AWS CCFs. This suggests that the ERA-Interim output provides a good representation of the occurrence of the different wind states, but the magnitudes from the ERA-Interim underestimate the values observed by AWS. This indicates that ERA-Interim output would not be a reliable way to identify relevant wind thresholds used in models

simulating polynya dynamics. This factor makes the usage of ERA-Interim problematic for polynya studies as they generally form on coastlines near topography in Antarctica.

Sea ice motion vector anomaly composites indicate that wind driven sea ice drift is significant 12 hours before through to 36 hours after strong wind events peak. This coincides with the peak cross correlation between SIC and wind speed, indicating



that during a strong wind event, SIC within the RSP is at a minimum during the period where wind driven sea ice drift is found to be significant. Following strong wind events, significant negative SIC anomalies are also found and persist for up to 5 days after the event. This period is longer than the persistence of most strong wind events (see autocorrelation in figure 3 and 4), it therefore seems unlikely that the SIC recovery following strong wind events is controlled by wind driven processes

such as sea ice drift. Thus, the recovery of sea ice is likely dominated by thermodynamic processes rather than dynamic processes, an idea also supported by recent analysis detailed in Nakata et al. (2015). During periods of high wind, negative SIC anomalies were found within the Ross Sea Polynya. Similarly positive anomalies were found during periods of low wind. The significant anomalies were also only found within areas of known polynyas, likely due to thinner, less concentrated sea ice being present within the RSP. Correlations between wind speed and SIC were significantly stronger for the high wind class

than the two weaker classes (Fig. 4), indicating that the wind driven polynya mechanism is driven by the strongest wind speeds and moderate winds have a less significant effect. This is likely due to the force applied to the sea ice being proportional to the wind stress or the wind speed squared theoretically.

    Only the results obtained from weather data taken at Laurie II are presented in this study because of its proximity to the RSP. However, a similar set of analyses were performed for the Vito, Emilia, and Ferrell AWS sites. These each produced similar

results, with the only significant difference being that weaker correlations were found at these sites when compared to those with Laurie II. This is hypothesised to be because these sites are based further inland, and are therefore more distant from the centre of the RSP than Laurie II. This would mean that the winds at these locations would be less representative than those at Laurie II.

    Only wind speed was considered in this study, so northerly winds were not distinguished from southerly winds. Even though

they would have a vastly different effect on polynyas. Due to the lack of topography on the Ross Ice Shelf, the dominance of katabatic fed drainage and barrier flows related to the Trans-Antarctic Mountains and a semi-persistent low pressure system east of the Ross Sea (Coggins et al., 2014) the vast majority of wind measurements in the north western RIS fall within the south-western quadrant (demonstrated by the wind roses in figure 1). This was also identified in detail analysis around the Laurie II region presented in Jolly et al. (2015 (in review). This directional bias becomes even stronger when only the largest

wind speeds are considered. This means that the effect of northerly winds on polynyas is minimal, and would not change our conclusions significantly.

    The sea ice motion vectors used in this study were not able to be derived accurately within 50 km of the coastline. This coastal area coincides with the majority of polynya activity and therefore the dynamic effects of changing wind speeds was not able to be observed directly within the RSP. The assumption is made that offshore sea ice drift will be representative of drift

within coastal polynyas. Although sea ice motion is coherent throughout the Ross Sea the motion within coastal polynyas may differ from offshore motion as thinner, less concentrated ice exists within coastal polynyas.

    The Bootstrap SIC data used throughout this study uses passive microwave measurements to calculate SIC. The microwave signature for a thin sheet of ice can be identical to that of scattered thick ice. For sea ice thickness less than 10 cm the Bootstrap sea ice concentration is a function of sea ice thickness Kwok et al. (2007). During periods of low wind speed, Bootstrap SIC

within the RSP often reaches 100% indicating a continuous covering sheet of sea ice with thickness greater than 10 cm. During



a strong wind event, the SIC decreases via dynamic processes leaving areas of open water with scattered, likely thick ice. As freezing of the open water occurs a layer of thin ice will form, causing a gradual increase in the Bootstrap SIC. As this sea ice thickens the heat flux between the warmer ocean and the cooler atmosphere will decrease, causing the rate of freezing to also decrease. Because both Bootstrap SIC and the rate of freezing within a polynya depend on the thermal conductivity of the sea

ice, the Bootstrap SIC may actually provide a more meaningful measure of sea ice production within polynyas than true SIC values.

Although Bootstrap SIC covers a vast period of time, we were unable to identify any significant trends in polynya activity. This is due to changing polynya structure as icebergs calve from the northern edge of the Ross Ice Shelf and the shelf gradually moves northward. As the coastline evolves, so does the RSP, this causes issues as the land mask for Bootstrap SIC data does

not change with time. Over long periods a changing amount of the RSP is visible in the Bootstrap SIC data, causing biases in any metric for polynya activity. This effect was particularly obvious in 2005 when iceberg B15 calved from the Ross Ice Shelf, moving a 300 km long section of the northern coastline 40 km further south.

## 5   Conclusions

During the austral winter, strong negative correlations were found between AWS wind speeds and SIC in the RSP. These

correlations persisted for several days and exceeded the persistence of the wind speed autocorrelation. When the data was split into low, medium and high wind cases and correlations were calculated from the separate data sets the high wind states displayed stronger correlations than the other two states indicating that stronger winds had the most significant impact on sea ice within the RSP. This analysis was repeated using a virtual AWS site interpolated from ERA-Interim reanalysis wind fields. It was found that although strong correlations existed between the AWS and ERA-Interim wind speeds, a linear scale factor,

significantly greater than 1 was present at AWS sites in close proximity to topography. Wind speeds measured at the Laurie II AWS correlated strongly with co-located ERA-Interim wind speeds, but a scale factor of 1.70 (indicating AWS wind speeds were 1.70 times faster than ERA-Interim wind speeds) was found. The ERA-Interim wind speeds were used to calculate a wind speed, SIC CCF that agreed with the CCF formed using AWS wind speeds. However, when the data set was categorised into low, medium and high wind cases and individual CCF's were calculated, significant differences from the AWS CCF's were

found in the high wind state. Due to the effects of nearby small scale topography, ERA-Interim wind speeds were unable to reproduce the relationships found between AWS wind speeds and SIC in the RSP.

For low, medium and high wind states measured at Laurie II AWS composites of SIC were made. During periods of low wind speed ($< 3.5 \ ms^{-1}$), significant, positive SIC anomalies that persisted for 5 days after the wind event were found within the RSP. Conversely, significant negative SIC anomalies that persisted for 5 days after the wind event were found within the RSP

during periods of high wind speed ($> 7.5 \ ms^{-1}$). No significant anomalies were found during medium wind speed periods. Significant SIC anomalies were only found to occur in coastal areas where polynya are known to occur. Composites of sea ice motion vectors were also calculated for these wind states. During periods of high wind anticyclonic motion anomalies were found throughout the Ross Sea, while cyclonic anomalies were found during periods of low wind. These sea ice motion



anomalies were found to persist for 48 hours after the wind event. SIC anomalies persisted several days longer than sea ice motion anomalies suggesting that following a strong wind event polynyas reform SIC through thermodynamic, rather than advective processes, resulting in increased sea ice production within polynyas following strong wind events.

*Acknowledgements.* We would like to thank the support of the University of Wisconsin-Madison Automatic Weather Station Program for

5   the AWS observational dataset (NSF grant numbers ANT-0944018, ANT-1245663, ANT-0943952, and ANT-1245737). We would also like to acknowledge the NSIDC for the provision of the SSM/I dataset. This work was partially funded by a grant from the New Zealand Antarctic Research Institute, scholarships from both the Department of Physics and Astronomy at the University of Canterbury and NZ Post (administered by Antarctica New Zealand).





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





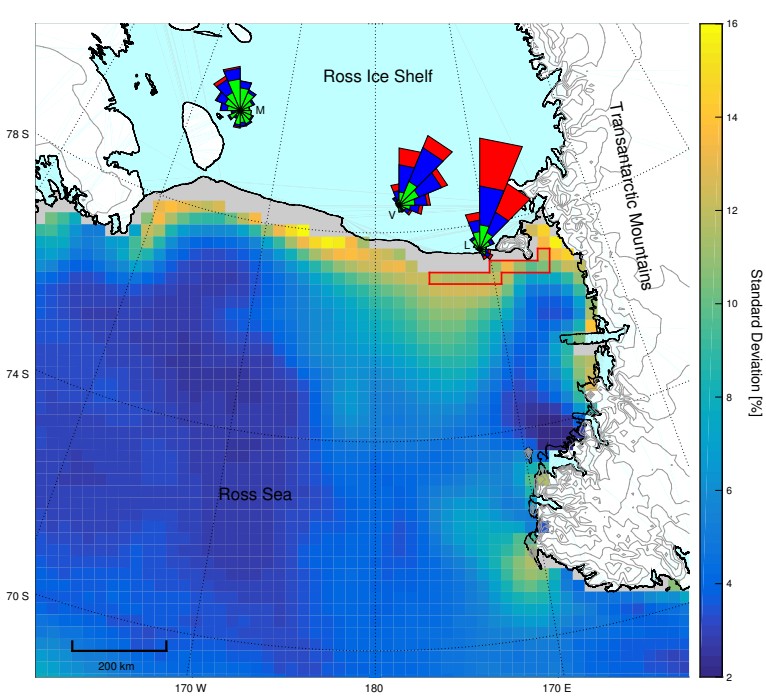

**Figure 1.** Map of Ross Sea Region, grey contour lines are spaced at 500 meters. The colour scale indicates the standard deviation of Bootstrap SIC for 20th April - 1st November, 1979 - 2014. Wind roses for Margaret, Vito and Laurie II AWS sites, labeled M, V and L respectively are included. The green, blue and red colours indicate the low medium and high wind speeds based on the 33rd and 66th percentile wind speeds (3.5 and 7.5 $ms^{-1}$) as measured at Laurie II.





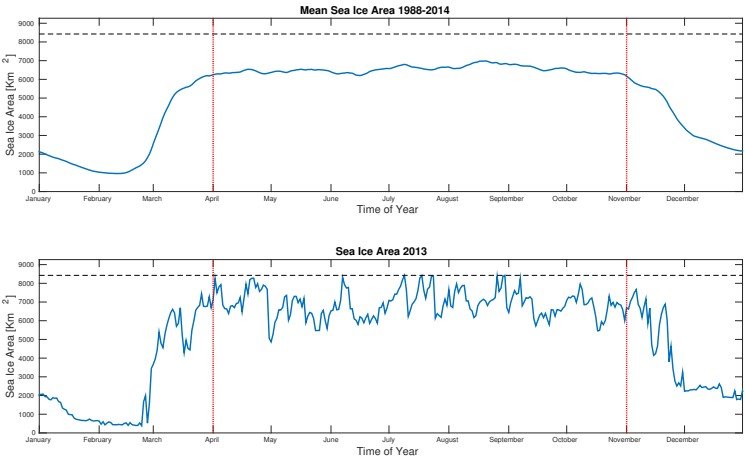

**Figure 2.** (a) Mean sea ice area within region north of Laurie AWS station. (b) Sea ice area during 2013, highlighting day to day variability. Red dotted lines identify the period between 1st April and 1st November used in this study, the grey dashed line indicates the maximum sea ice area possible within the region defined in Figure 1.

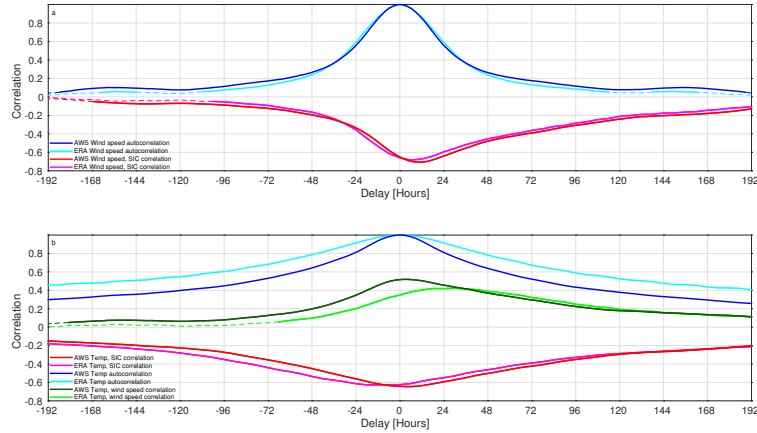

**Figure 3.** (a) Cross correlation curve for sea ice concentration and AWS (red) and ERA-Interim (magenta), wind speeds and autocorrelation curve for wind speed at Laurie II AWS (blue) and ERA-Interim (cyan). (b) Cross correlation curve for sea ice concentration and temperature (red) and ERA-Interim (magenta), autocorrelation curve for wind speed at Laurie II AWS site (blue) and ERA-Interim (cyan). Cross correlation for wind speed and temperature measured at Laurie II AWS (dark green) and ERA-Interim (light green). Dashed lines indicate significance $p > 0.01$.





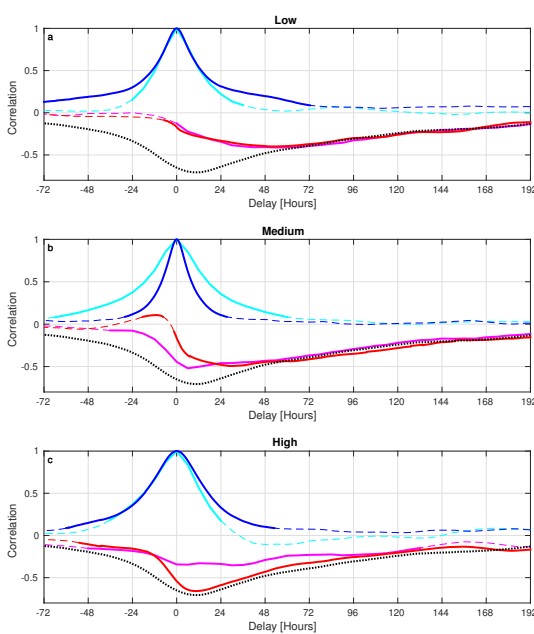

**Figure 4.** Cross correlation curves for SIC and AWS (red) and ERA-Interim (magenta) wind speeds at Laurie II AWS site for low (a), medium (b) and high (c) wind cases. Autocorrelation curves for low medium and high wind cases from AWS (blue) and ERA-Interim (cyan). Dashed lines indicate significance $p > 0.01$. The dotted, black line represents the cross correlation curve for the total of all wind cases for comparison.



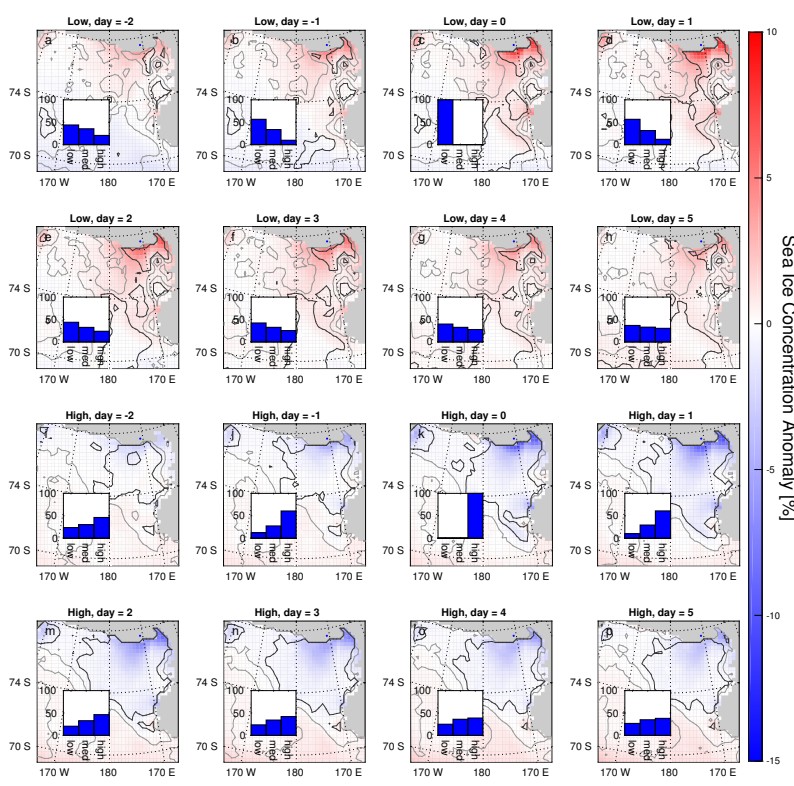

**Figure 5.** Sea ice concentration anomalies at varying delay for low wind cases (a-h) and high wind cases (i-p). The grey and black contours indcate 80% and 99% significance respectively. Inset histograms indicate the percentage of the three wind cases that occur at that delay.





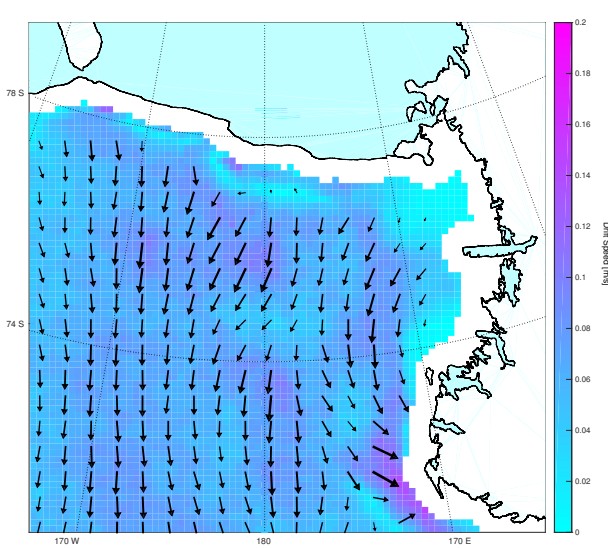

**Figure 6.** Mean sea ice motion vectors in the Ross Sea region.





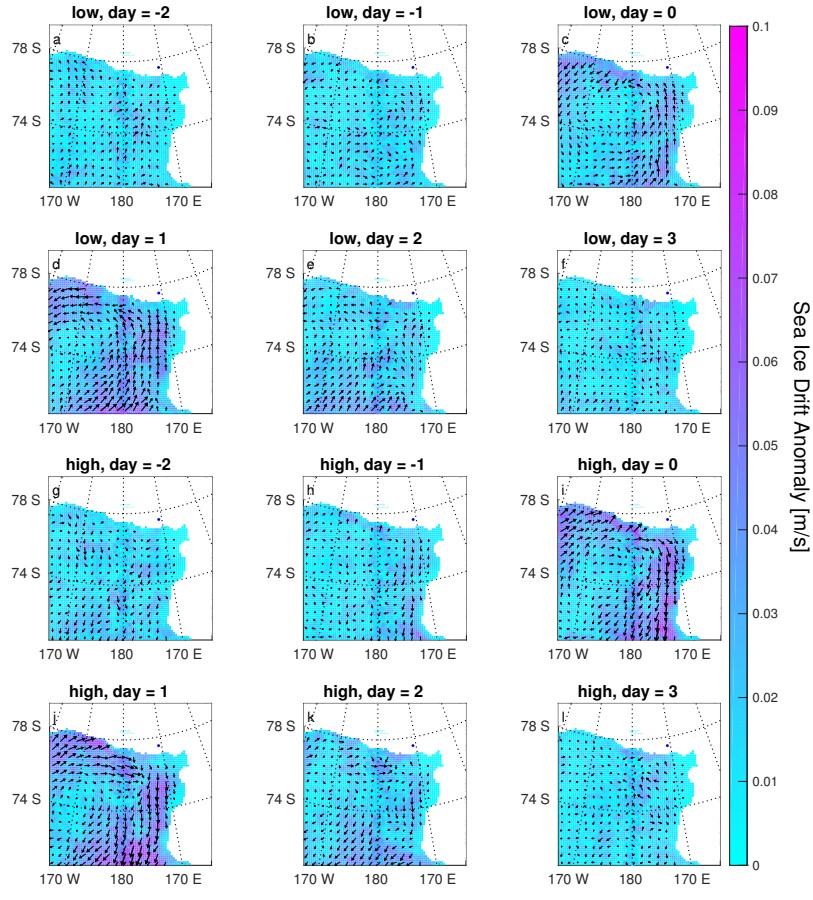

**Figure 7.** (a-f) Sea ice motion anomalies for days -2 to 3 for low wind events. (g-l) Sea ice motion anomalies for days -2 to 3 for high wind events.