# Peer review of "Atmospheric forcing of sea ice anomalies in the Ross Sea Polynya region"

_The Cryosphere, 2016_

## Referee Comment (RC1) · Anonymous Referee #1 · 14 Jun 2016

General Comments:

The authors examined the correlations between SIC, AWS winds/temp, ERA-Interim winds within a defined region in Ross Sea Polynya. They found persistent weak/strong winds near the edge of the Ross Ice Shelf are generally associated with positive/negative SIC anomalies in the Ross Sea Polynya (RSP). They also report rapid decreases in SIC during a strong wind event are followed by a more gradual recovery in SIC perhaps due to the slower responses of thermodynamic processes to changes. Comparison of AWS and ERA-I winds also suggests that ERA-I winds are weaker than observed.

There are a lot of details in the manuscript, some of which are interesting and some requires clarification (Please see comments and queries below) and justification based

on the quality of the SIC retrievals. The quality of the SIC may be confounding in some cases and may affect (and therefore not support) the conclusions about the process behavior. Please address the concerns below.

My recommendation is for publication after revisions.

More specific comments:

Page.line

Abstract. This would be more effective if shortened in length. The first paragraph seems out of place in an abstract.

Fig. 1 Please define the red polygon/region in the caption, as you have described in text (or refer the reader to the text).

4.25 I have to ask: Even though different investigators have derived ice motion from the passive microwave data set using similar methodology, their quality varies. How have you assessed your derived ice motion estimates?

Fig. 3 caption: (b) red is the AWS and magenta is ERA-Interim.

5.25 I assume 2-meter winds are used in these analyses. Otherwise, there would be a scale factor. Also of interest is whether the ERA-I winds are directionally biased.

6.10 The Bootstrap algorithm is based on binned TBs over a day, so there is a blurring of events (polynya openings) over a 24-hour period. Please clarify the sentence re:varying time lag in 6.5.

6.15 Isn't this also dependent on the response of the Bootstrap retrievals to changes in observed brightness temperature?

6.18 There is lag between the changes in wind direction observed at Laurie and at the RSP?

6.28 Your arguments re:lag seem reasonable, but I'm still not quite comfortable as to

whether the 24-hr sampling of the SIC fields would support your attribution statements. Perhaps I still don't quite clear about your remarks in 6.5.

6.29 You mean the wind speeds autocorrelation has e-folding time of 36 hours. If you included direction, it may be different.

7.10 You should also note that this also depends on the response of the bootstrap algorithm to thin ice growth. The algorithm designates thin ice as open water until the ice reaches a thickness of about 20 cm. So, that may explain some of the asymmetry in the responses.

8.0 At this point, I recommend that the results section should be broken into subsections. As is, there are five pages of text.

8.15 A general question: Are they larger differences between the AWS and ERA-I winds when the winds are strong (e.g., katabatics).

9.6 OK, these anomalies are interesting. I guess this is presented as just a remark on the results?

11.10 I thought Bootstrap accounted for the changing coastline.

---

## Referee Comment (RC2) · Anonymous Referee #2 · 2 Jul 2016

General Comments:

This is a comprehensive analysis of the "winter" sea ice concentration behavior in the Ross Sea polynya in relation to the atmospheric forcing. The near-surface atmospheric conditions affecting the polynya are approximated by observations of 3-m winds and temperatures from the Laurie II AWS, just to the east of Ross Island and adjacent to the northern edge of the Ross Ice Shelf. The results are very consistent with earlier work showing the major impact of strong wind events. In fact, this analysis has a direct predecessor in work of Bromwich et al. (1998) who performed a limited investigation of the Ross Sea polynya variations in relation to winds and temperatures from the Ferrell AWS, yet this paper is barely mentioned. A comparison with the ERA-Interim global reanalysis fields is also included, and shows many similarities with the AWS-based analysis. There are a number of issues that should be addressed before publication.

Specific Issues:

1. An explicit discussion should be provided as to how the wind and temperature observations at Laurie AWS are expected to differ from those over the Ross Sea polynya to the north. This should be based on topography, roughness length, and stability differences. 2. At several locations in the manuscript (e.g., pages 2 and 5) reference is made to the unavailable Jolly et al. (2015) manuscript to claim that the Antarctic Mesoscale Prediction System is unable to resolve the main topographically forced winds of relevance to the behavior of the Ross Sea polynya, therefore implicitly justifying the use of ERA-Interim instead. Given model grid spacings for the Ross Island region as fine as 1.1 km and the generally favorable AMPS validation studies (e.g., Bromwich et al. 2005, Monthly Weather Review), this claim is quite surprising and requires expanded discussion and substantiation. 3. Given the rather similar analysis by Bromwich et al. (1998), much more consideration and contrasting its results with those here is required. 4. In the conclusions section, a major effort should be made to compare and contrast the present findings with those from earlier work, i.e., to address the issue of "what new aspects have been revealed by the present effort"? 5. How is the standard deviation of SIC shown in Fig. 1 actually calculated? Is this based on the daily variation from the interannual mean of SIC for the same day? Why start on April 20 rather than April1? 6. Reference is made to the "winter" being April – November. It is more accurate to give the period as April-October. 7. It would be helpful to keep reminding the reader why you are using the 2001-2014 period, probably because of Laurie AWS data availability. 8. Fig. 1: An inset map is needed showing the Ross Island topography and the locations of Laurie, Ferrell, and Emilia AWS. 9. Fig. 2: Why is the maximum sea ice area exceeded in 2013? 10. Fig. 3: Specify how positive and negative delays are defined, i.e., which variable leads? 11. Fig. 5: I didn't understand the construction and meaning attached to the histograms. 12. Page 8, lines 5-7: This sentence needs to be more carefully formulated for accuracy and understandability. 13. Page 9, lines 14-16: How do incoherent motions demonstrate the critical influence of surface winds on sea ice motion? 14. Page 11, lines 32-33: This sentence is backwards – high winds are

associated with cyclonic motion anomalies.

Technical Issues: 1. Page 4, line 10: "Margaret" – typo. 2. Page 5, line 2: "water vapor"? Line 30: "System" - typo. 3. Page 6, lines 29-32: Rewrite this sentence for clarity. 4. Fig. 5 caption: "indicate" – typo.
* * *

---

## Author Comment (AC1) · 8 Aug 2016

We thank the reviewer for taking the time to provide constructive comments on our manuscript. We feel the comments provided have allowed us to significantly improve the clarity and structure of our text. Response to specific comments follow, with reviewers comments quoted in bold.

**Abstract. This would be more effective if shortened in length. The first paragraph seems out of place in an abstract.**

The Abstract has been edited, it now reads:

"The impacts of strong wind events on the Ross Sea Polynya (RSP) and its sea ice concentration and possible consequences for sea ice production are investigated. We utilised bootstrap sea ice concentration (SIC) measurements derived from satellite SSM/I brightness temperatures and compared these with surface winds and temperatures from automatic weather stations (AWS) and weather models (ERA-Interim). Daily data in the austral winter period were used to classify characteristic weather regimes based on the percentiles of wind speed. For each regime a composite of a SIC anomaly was formed for the entire Ross Sea region and we found that persistent weak winds near the edge of the Ross Ice Shelf are generally associated with positive SIC anomalies in the Ross Sea Polynya and vice versa. By analysing sea ice motion vectors derived from the SSM/I brightness temperatures we find during strong wind events significant sea ice motion anomalies throughout the Ross Sea which persist for several days after a strong wind event has ended. Strong, negative correlations are found between SIC and AWS wind speed within the RSP indicating that strong winds cause significant advection of sea ice in the region. We were able to partially recreate these correlations using co-located modelled ERA-Interim wind speeds. However, large AWS and model differences are observed in the vicinity of Ross Island, where ERA-Interim underestimates wind speeds by a factor of 1.7 resulting in a significant misrepresentation of RSP processes in this area based on model data. Thus, the cross correlation functions produced by compositing based on ERA-Interim wind speeds differed significantly from those produced with AWS wind speeds. In general the rapid decrease in SIC during a strong wind event is followed by a more gradual recovery in SIC. The SIC recovery continues over a time period greater than the average persistence of strong wind events as well as the persistence of sea ice motion anomalies. This suggests that the production of new sea ice occurs through thermodynamic rather than dynamic processes."

**Fig. 1 Please define the red polygon/region in the caption, as you have described in text (or refer the reader to the text).**

Added "The red line outline indicates the region discussed in Section 2." to the caption.

**4.25 I have to ask: Even though different investigators have derived ice motion from the passive microwave data set using similar methodology, their quality varies. How have you assessed your derived ice motion estimates?**

No validation was made, however we used a standard technique (Emery et al. 1997, Heil et al. 2006, Holland and Kwok 2012). The results show a mean value comparable to other authors (Emery et al. 1997, Heil et al. 2006, Holland and Kwok 2012) and Figures 6 and 7 show physically interpretable signatures this supports the measures used.

**Fig. 3 caption: (b) red is the AWS and magenta is ERA-Interim.**

This error has been corrected.

**5.25 I assume 2-meter winds are used in these analyses. Otherwise, there would be a scale factor. Also of interest is whether the ERA-I winds are directionally biased.**

Scale factors have not been used, We are primarily interested in correlations which will remain unaffected. For the intercomparison this will cause a small difference in the observed gradient between the two datasets. AWS heights are consistently set to 2m or 3m (Lazzara et al., 2012) while ERA-I winds are 10m and this difference cannot explain the large variation in the gradients found. The following was added to the discussion:

"ERA-Interim provides 10 m wind speeds while the AWS Wind speeds are measured at 2-3 m (Lazzara et al., 2012, Dee et al. 2011). This would suggest that a scale factor would exist between the two data sets, an effect that was not corrected. While this will not affect the correlation comparisons preformed it may explain the scale factors observed. However this small height difference is not able to explain the large scale factors found indicating that topography must have a significant effect."

Inspection of the wind roses for both AWS and ERA-I winds at Laurie II show no obvious directional bias (Fig 1). Sentence has been added.

"Inspection of ERA-Interim and AWS wind roses revealed no significant directional bias between the two data sets.".

**6.10 The Bootstrap algorithm is based on binned TBs over a day, so there is a blur- ring of events (polynya openings) over a 24-hour period. Please clarify the sentence re:varying time lag in 6.5.**

The blurring due to the temporal resolution of Bootstrap is a limitation of our analysis. CCF's were produced from daily bootstrap measurements and 24 hour rolling means of 10 min resolution AWS data. This allowed CCF's to be produced at 10 min resolution. The CCF's produced in this manner will be subject to blurring from both the Bootstrap algorithm and the 24 hour means of wind speed used, and are therefore oversampled. The original manuscript justifies the significant correlation at negative delay between AWS wind speeds and SIC as the AWS wind speed autocorrelation has a similar e-folding time to that of the CCF (6.30). The e-folding time for both the CCF and the autocorrelation is limited by this blurring. The sentence "The Bootstrap SIC is derived from 24 hour binned brightness temperatures and we compared these with 24 hour rolling mean of AWS data the resulting correlation functions will be blurred over a 24 hour period." was added to the results section to acknowledge this limitation.

[Figure]

**6.15 Isn't this also dependent on the response of the Bootstrap retrievals to changes in observed brightness temperature?**

The decay rate here is impacted by temporal resolution of bootstrap data, but the decay rate of the autocorrelation is greater than the 24 hour blurring period so other factors must contribute.

**6.18 There is lag between the changes in wind direction observed at Laurie and at the RSP?**

We have not focussed on wind direction changes which are beyond the scope of the analysis because of the lack of data within the RSP. Also wind direction is rather persistent and therefore likely has less impact.

**6.28 Your arguments re:lag seem reasonable, but I'm still not quite comfortable as to whether the 24-hr sampling of the SIC fields would support your attribution statements. Perhaps I still don't quite clear about your remarks in 6.5.**

See previous clarification regarding 6.10.

**6.29 You mean the wind speeds autocorrelation has e-folding time of 36 hours. If you included direction, it may be different.**

This error has been corrected.

**7.10 You should also note that this also depends on the response of the boot-strap algorithm to thin ice growth. The algorithm designates thin ice as open water until the ice reaches a thickness of about 20 cm. So, that may explain some of the asymmetry in the responses.**

Kwok et. al. 2007, compares Nasa Team 2 and bootstrap SIC retrievals with AMSR-E derived sea ice thickness. Kwok et al. (2007) idnetified: "From the monthly analyses an ice thickness of $10cm$ corresponds to $83 \pm 3\%$ and $91 \pm 2\%$ ice concentration in the ABA [Bootstrap] and NT2 [Nasa Team 2] estimates." The effect of this is discussed in the original manuscript at 10.32.

**8.0 At this point, I recommend that the results section should be broken into subsec- tions. As is, there are five pages of text.**

This change has been made.

**8.15 A general question: Are they larger differences between the AWS and ERA-I winds when the winds are strong (e.g., katabatics).**

Figure 2 indicates a linear relationship between AWS and ERA-I wind speeds with slope of 1.70 as discussed at 5.24 of the original manuscript. Larger absolute differences between AWS and ERA-I are observed at high wind speeds, although the relationship is linear.

**9.6 OK, these anomalies are interesting. I guess this is presented as just a remark on the results?**

We agree with the reviewer, but have left this as is based on discussion between co-authors.

**11.10 I thought Bootstrap accounted for the changing coastline**.

The NSIDC product uses a fixed land mask. Comiso et al. (2011) discuss the difficulties producing a dynamic land mask. We do not consider trends over the time period and this will therefore not impact our results.
* * *
[Figure]

[Figure]

**Fig. 1.** Wind roses for winds at Laurie II for AWS (left) and ERA-I (right) winds. The green indicates wind speed less than 3.5 m/s, blue between 3.5m/s and 7.5m/s and red greater than 7.5m/s.

[Figure]

[Figure]

**Fig. 2.** Scatter plot of winds at Laurie II for AWS and ERA-I wind speeds. The red line indicates the linear fit.

---

## Author Comment (AC2) · 8 Aug 2016

We would like to thank the reviewer for providing useful comments on the text. These have allowed us to strengthen our manuscript and increase the clarity of our discussions. Response to specific comments follow, with reviewers comments quoted in bold.

**1. An explicit discussion should be provided as to how the wind and temperature observations at Laurie AWS are expected to differ from those over the Ross Sea polynya to the north. This should be based on topography, roughness length, and stability differences.**

Due to lack of weather data available within the RSP, AWS data from sites on the nearby

[Figure]

Ross Ice Shelf was assumed to be representative of that over the RSP. A multitude of effects will cause the actual winds and temperatures over the RSP differ somewhat from that measured at the Laurie II AWS site, these effects will be inhomogeneous across the RSP. Proximity to topography, particularly Ross Island differs between the RSP and Laurie II. The majority of the RSP is more distant to Ross Island than Laurie II, but some areas of the polynya will be closer as it reaches the northern coastline of Ross Island. Southerly katabatic drainage flows will accelerate around Ross Island, so the net result of Ross Island on these winds at Laurie II will be to cause greater wind speeds. This effect will be present in some areas of the RSP but others will be too distant for this to be significant. Areas of the RSP north of Ross Island will be somewhat sheltered from many of the predominant southerly winds. The surface roughness of the Ross ice shelf, beneath Laurie II AWS will differ to that within the RSP. This will depend on the current state of the RSP. This difference will cause the winds within the boundary layer to differ between these locations. Due to the relatively warm ocean an upward heat flux will occur within the RSP when open water or thin ice is present. This will cause an increase in surface air temperatures over the RSP. This effect will not occur at Laurie II due to the insulation of the thick ice shelf. Due to the lack of measurements within the RSP the magnitude of these effects is unable to be identified.

A shortened summary of this has been added to the discussion:

"Due to lack of weather data available within the RSP, AWS data from sites on the nearby Ross Ice Shelf was assumed to be representative of that over the RSP. A multitude of effects will cause the actual winds and temperatures over the RSP differ somewhat from that measured at the Laurie II AWS site, these effects will be inhomogeneous across the RSP. Proximity to topography, particularly Ross Island differs between the RSP and Laurie II. Southerly katabatic drainage flows will accelerate around Ross Island, causing stronger winds to be observed within the RSP. Meanwhile other areas of the RSP north of Ross Island will be somewhat sheltered from many of

the predominant southerly winds. Due to the relatively warm ocean an upward heat flux will occur within the RSP when open water or thin ice is present. This will cause an increase in surface air temperatures over the RSP. This effect will not occur at Laurie II due to the insulation of the thick ice shelf. Due to the lack of measurements within the RSP the net result of these effects is unable to be identified."

**2. At several locations in the manuscript (e.g., pages 2 and 5) reference is made to the unavailable Jolly et al. (2015) manuscript to claim that the Antarctic Mesoscale Prediction System is unable to resolve the main topographically forced winds of relevance to the behavior of the Ross Sea polynya, therefore implicitly justifying the use of ERA-Interim instead. Given model grid spacings for the Ross Island region as fine as 1.1 km and the generally favorable AMPS validation studies (e.g., Bromwich et al. 2005, Monthly Weather Review), this claim is quite surprising and requires expanded discussion and substantiation.**

This paper is now available at *http://journals.ametsoc.org/doi/abs/10.1175/MWR-D-15-0447.1*. We did not use ERA over AMPS because of the issues highlighted in Jolly, but rather because of the temporal consistency of the ERA dataset. AMPS is not a consistent product, since it is operational, making it difficult to use for a longer term climatological study. AMPS also began in 2002 which would cause issues with its usage in this application. A sentence justifying the use of ERA over AMPS has been added:

"Although AMPS provides higher resolution weather data, ERA-Interim was used due to its temporal consistency and data set spanning the AWS period."

**3. Given the rather similar analysis by Bromwich et al. (1998), much more consideration and contrasting its results with those here is required.**

We feel that our results add to the Bromwich et al. (1998) work significantly, for example the usage of lags to probe physical processes. However, we have added the following material comparing this work with those previous results.

"Bromwich et al. (1998) found annual correlations between SIC in the RSP and wind speed at Ferrell AWS for 1988-1991 ranging from -0.3 to -0.52. We find the multi-year correlation for SIC in the RSP and wind speed at Ferrell AWS from 2001 to 2014 to be -0.67. The disagreement between these values is to be expected as Bromwich et al. (1998) uses a RSP area that extends significantly further from the shore than the one used in our analysis. Winds over their RSP area are not as well represented by the Ferrell AWS as the area used within our analysis justifying the weaker correlation observed. We find a minimum correlation between Ferrell wind speed and SIC to be -0.72 at 10 hours delay, Bromwich et al. (1998) did not calculate correlations at varying delay so comparison with this value is not possible. Bromwich et al. (1998) also find correlations between SIC and inverse temperature ranging between 0.44 and 0.55. We found a SIC, inverse temperature correlation of 0.639 this difference is due to the different RSP areas used."

**4. In the conclusions section, a major effort should be made to compare and contrast the present findings with those from earlier work, i.e., to address the issue of "what new aspects have been revealed by the present effort"?**

This is covered by point 3.

**5. How is the standard deviation of SIC shown in Fig. 1 actually calculated? Is this based on the daily variation from the interannual mean of SIC for the same day? Why start on April 20 rather than April1?**

The standard deviations are the daily variations from interanual mean SIC for the entire winter (in this case 20 April-1 November). The 20th of April was used as the start of the winter here as winter sea ice had not formed in the northern areas of the map until this time. Using the first of April results in high standard deviation values in northern areas of the map. These high standard deviations are due to the seasonal cycle not polynya activity so the beginning of April was removed to make the result clearer for the reader. The first of April was used elsewhere to maximise the size of the data set as all other analysis was preformed near the edge of the ice shelf, where sea ice forms early in the season. The definition of the standard deviation has been made clearer and an explanation of the varying time periods used has been added. The beginning of paragraph 2 of data and methods section now reads;

"Fig. 1 shows the standard deviation of SIC. This is defined as the daily variation from the inter-annual winter mean of Bootstrap SIC over the period, 20th April until the 1st of November for years 1979 until 2014. This period was chosen to exclude the annual break out of sea ice which would add variability not associated with day to day polynya activity"

**6. Reference is made to the "winter" being April – November. It is more accurate to give the period as April-October.**

Agreed, this error has been corrected.

**7. It would be helpful to keep reminding the reader why you are using the 2001-2014 period, probably because of Laurie AWS data availability.**

Your assumption is correct a reminder has been added at Page 9 Line 5.

**8. Fig. 1: An inset map is needed showing the Ross Island topography and the locations of Laurie, Ferrell, and Emilia AWS.**

Room for a inset within this figure is scarce, and the figure already contains a lot of information. The co-author group did not feel that a standalone figure was justified.

**9. Fig. 2: Why is the maximum sea ice area exceeded in 2013?**

This appears to be a graphical error due to the thickness of the line, the centre of the line never crosses $100\%$. The line thickness's have been modified to minimise this effect

**10. Fig. 3: Specify how positive and negative delays are defined, i.e., which variable leads?**

Added definition of delay to caption;

"The delay is defined such that positive indicates meteorology measures leading SIC.".

**11. Fig. 5: I didn't understand the construction and meaning attached to the histograms.**

Added line in text to clarify histograms;

"Histograms are also shown to indicate how the distribution of each wind class changes throughout the period examined. On day 0 all cases are either 100% high

or low winds, but on following days the winds are not classified, this allows the persistence of these wind events to be observed."

**12. Page 8, lines 5-7: This sentence needs to be more carefully formulated for accuracy and understandability.**

Text was changed;

"All cases show negative correlations between SIC and wind speed. With the exception of a short period in the medium case, spanning -30 hours to 6 hours when weak positive correlations are observed."

**13. Page 9, lines 14-16: How do incoherent motions demonstrate the critical influence of surface winds on sea ice motion?**

This has been changed to:

"It is also noticeable that no coherent pattern in the sea ice anomalies associated with the medium wind state are observed (not shown). The cyclonic anomalies during strong wind events and anticyclonic anomalies during low wind events highlight the critical influence of atmospheric near-surface winds on sea ice motion in the region."

**14. Page 11, lines 32-33: This sentence is backwards – high winds are associated with cyclonic motion anomalies.**

This error has been corrected.

**Technical Issues: 1. Page 4, line 10: "Margaret" – typo. 2. Page 5, line 2: "water vapor"? Line 30: "System" - typo. 3. Page 6, lines 29-32: Rewrite this sentence for clarity. 4. Fig. 5 caption: "indicate" – typo.**

These errors have been corrected.

---

## Author Response (AR2)

We would like to thank the reviewers for taking the time to provide feedback on the manuscript. The feedback provided has helped significantly improve the structure and flow of the manuscript.

Changes are indicated in the latexdiff that follows. All line numbers quoted here refer to the latexdiff document in the form "page.line". The reviewers comments follow in bold.

**Response to Editor's comments**

**thank you for the revisions of your manuscript. As the reviewers had wished to see it again, I will send it to them for further review. However, in the meantime, I would be glad if you would consider to also include the figures from your reply (at least Figure 2) in the actual manuscript, to support some of your conclusions.**

Figure 2 has been added.

**Also, I find the captions of Figures 5 and 6 confusing. Could you please clarify/structure better.**

Captions have been edited.

**Why are the ACFs in Figure 7 asymmetric?**

We assume you refer to figure 4 here as figure 7 does not contain ACFs. Splitting the data into low, medium and high wind speeds results in a data set that is not continuous. This sampling issue allows differences to occur between positive and negative lag. These differences are larger for ERA-Interim due to the lower temporal resolution of this data set. Changed sentence at 9.16.

**Response to Reviewer 1**

**I am satisfied with the revisions. I would like to request one revision in the abstract: The following statement is particularly jarring and should be reworded: "...This suggests that the production of new sea ice occurs through thermodynamic rather than dynamic processes..." Perhaps I'm being picky but isn't new ice production always thermodynamically forced and never dynamically forced?**

This sentence was also found troublesome by Reviewer 3, the sentence has been edited (1.19).

**Response to Reviewer 3**

**Recommendations: This study addresses an important and relevant issue and is in principal suitable for publication in The Cryosphere. While the methodology, analysis, and conclusions seem mostly robust, I have some concerns regarding the presentation and structure of the manuscript. I found it very difficult to read and to follow the line of thought, which should be improved alongside some other concerns listed below before I would recommend the publication of the manuscript. A general advice would be to organize and use paragraphs in a more structured sense, i.e., to have one topic**

per paragraph with the principal idea of the paragraph expressed in the first sentence and the concluding statement in the last sentence. Moreover, it would be easier to read if the paragraphs would neither be only 2-3 sentences long nor a full page. I think in some places (see below) the text could be a bit more focused and concise and certainly there are still quite some typographical errors that require a careful check (due to temporal constraints I cannot list them here, please check).

The editor asked specifically whether I thought that the concerns by Referees #1 and #2 were sufficiently addressed. I think the authors did address these issues sufficiently in most cases. I listed some remaining issues and some additional concerns that should be addressed in a revision below.

**Major issues:**

**Introduction:**

**- Large parts of the introduction deal with the increase of Antarctic sea ice over recent decades, but I do not really see how this study contributes to this question. It might well be that polynya production and export in the Ross Sea changed in concert with the observed ice expansion in the Ross Sea (e.g. Drucker et al., 2011 and Haumann et al., 2016), but the causal link between the ice edge variation and the polynya variation is less well established. This study does neither study variations of the sea ice extent nor long-term trends. So, I would recommend to focus the on the more important aspects, which are local aspects and temporal variability. One of the aspects is e.g. the effect on deep water formation (e.g. Ohshima et al. 2013) or ecology.**

A large section of the introduction discussing global SIE trends has been removed (3.8-14). A small section discussing the effects of the RSP and deep water formation has been added at 2.24-27. A relevant sentence has also been added to 3.32.

**- The introduction could be better structured. Right now the reader is being put back and forth between sea-ice trends and polynya processes. It might be helpful to point out more clearly the gap of knowledge, which is the effect of the local wind system on the polynya processes that cannot be resolved by models and end on the contribution of this study.**

The Introduction has been significantly restructured, see previous comment.

**Data and methods:**

**- It is not fully clear to me which years are used in the end for the analysis. The authors list a number of different time periods over which the different data sets are available, but ultimately I guess the analysis period is constrained by the availability of the AWS data, which only existed after 2000. So, all data sets, analysis, figures should consistently build on this period and has to be mentioned somewhere.**

The 2000-2014 period used is more clearly stated (6.29).

**- Was the data de-trended and de-seasonalized before performing the analysis? I think it would change the results much, but it should be done when using a correlation analysis.**

The data was not de-seasonalised and de-trended before performing this analysis. As discussed in 5.27-29 the SIC during the period analysed is very constant and has no strong seasonal trend so we believe the correlation analysis done is still valid. This is supported by the CCF's decaying towards zero at large lag periods. If the correlations found were a result of a seasonal

effect this would not be the case.

**- The drift vectors are calculated over multiple grid boxes that have a size of 25 km and one vector is obtained from differencing multiple grid boxes. Therefore, a resolution of 25 km for the final motion product seems not adequate as the actual resolution is coarser than that. Ideal would be to provide the vectors at their actual resolution or describe this caveat.**

The drift vectors were calculated from 12.5 km SSMI grid boxes (4.33). Following the differencing the velocities found were then smoothed to 25 km resolution (5.8) to minimise oversampling. Furthermore the vectors plotted in figures 6 and 7 are smoothed further and plotted at 100 km resolution, while the colourmap remains at 25 km resolution. 100 km resolution stated in figure caption.

**- Throughout the manuscript the authors use sea-ice concentration (SIC) in the text but Figure 2 actually shows sea-ice area while the text says it would show SIC. It needs to be clarified what quantity is used when and why.**

Because we are considering a fixed region there is a direct proportionality between SIC and SIA. However, we agree that the use of SIA on Fig 2 is needlessly confusing therefore Figure 2 has been modified to use a SIC scale instead of sea ice area.

**- Lines 4.24-25: Please describe how you actually calculated the ice velocity from the cross-correlation field or provide a reference.**

Section describing this method has been edited for clarity (5.5-14) The referenced works (Emery et al., 1997; Heil et al., 2006; Holland and Kwok, 2012) also describe this method.

**Results:**

**- The splitting of the results section as suggested by one of the earlier reviewers definitely helped to improve the readability. However, some more structuring would be helpful. The first paragraph of section 3.3 should be split and restructured. I honestly have difficulties to follow and I think that some parts might be cut. For example, I found the explanation of lines 6.3-6 regarding the wind stress more confusing than helpful at this point.**

6.26-29 was moved to the discussion (12.1-3) where we feel it fits better. Section 3.3 has been separated into several paragraphs to improve readability.

**- Section 3.4 is supposed to be about ERA-Interim data (title) but in fact half of it is about the AWS data, which I found confusing. Please restructure to resolve this issue. Also I think the information in this section could be more confined and the second paragraph should be shortened, split or restructured for clarity.**

Title of section changed to "Comparison of ERA-Interim and AWS CCFs" which better reflects the content. lines 8.24-26 have also been removed.

**- I do not understand Figure 4 a. It shows a negative correlation between the wind speed and SIC at low wind speeds. However, figure 5, the text, abstract and conclusions say that at low wind speeds the ice concentration is higher. There-**

**fore, I would expect a positive correlation here. To me this seems inconsistent. Could you explain?**

The author team is not sure if we follow your logic here. A negative correlation between SIC and wind speed implies that during periods of low wind speed SIC will be higher than normal. This remains true even when we consider a subset of the data.

**Discussion:**

**- Lines 10.33-11.1: To me this formulation does not make sense. How could sea-ice drift form ice. It could only advect ice into this region, which would be the reversal of the opening process. However, this seems unlikely. On the other hand, the statement that "wind-driven processes" are not responsible for the recovery, seems unsupported to me as well. As there would be no recovery phase without the prior wind-driven off-shore advection.**

SIC recovery and sea ice formation are not necessarily the same. Sea ice recovery in the RSP can either occur through formation of new ice within the RSP or by southward advection of existing sea ice. Southward advection is unlikely as it opposes the dominant prevailing wind in the region. Sea Ice anomalies are found to persist beyond the autocorrelation period of winds so the SIC recovery cannot be directly driven by wind rather indirectly through sea ice production. Lines 11.26-27 have been reformulated for clarity.

**All figures:**

**Currently many of the labels, legends, and arrows are rather small. Please enlarge them for readability.**

Figures have been edited.

**Some further suggestions:**

**Lines 1.2-3: Delete or reformulate "and possible consequences for sea ice production". Sea ice production is not "investigated" in this study. I agree that the study has implications for the sea ice production but there is no related analysis.**

1.1-5 has been reformulated using sea ice formation in place of production which implies a process rather than a product.

**Lines 1.18-19: "This suggests …". I think that this sentence is slightly problematic, since the thermodynamic ice production mostly occurs because the ice was dynamically removed previously. So, it is not a pure thermodynamic process that leads to the polynya ice production. I agree that the ultimate new ice formation occurs due to thermodynamics, but isn't that the case by definition? Or in other words how could new ice be produced dynamically. Please reformulate.**

Change has been made at 1.19.

**Line 1.22: "growth" -> "expansion"**

Change has been made at 1.23.

**Lines 2.20-23: I do not see the link between the ice extent or these processes to this study or what was discussed in**

**this paragraph.**

Changed ice extent to SIC at line 2.34. These processes strongly influence the SIC within a polynya, the large number of processes that are controlled by various systems is the reason why polynyas are difficult to understand.

**Line 2.25: "... wind stress." This statement requires in my view a reference.**

Added reference to Holland and Kwok (2012). (2.17)

**Lines 2.29-31: SIE is not used in this study, please reformulate or remove.**

Lines (3.8-13) have been removed.

**Lines 3.7-17: Please be more precise on what you actually do in this study and remove or replace the rest for helping the reader what to expect from this study. Please take out the ice production, as this study does not analyze ice production.**

Changed to sea ice formation at line 3.30.

**Line 3.22: I think the record starts in "1978". Anyway, please only indicate the time period that you actually used and remove the information on the rest of the record.**

'1977' typo has been fixed at line 4.5.

**Lines 4.5-10: This seems to belong rather to the results section than Data and Methods.**

Section (4.22-26) has been moved to 6.4-8.

**Line 5.23: "Scale factor" -> Do you mean "slope of the regression line"?**

We agree that this wording is better than the way we described this factor and we have thus changed the corresponding text at 11.5

**Lines 6.14-15: Please be more specific on "dynamic" or "thermodynamic process", i.e. "northward advection", "ice formation", etc..**

The suggested changes were made at lines 7.8-9.

**Lines 6.24-28: As the sea ice is divergent in this situation, I doubt that ridging and rafting processes are responsible for the delay. I would think that it is rather the ice internal stress that might cause the delay in the ice response.**

The significance ridging and rafting was overemphasised in the previous manuscript. 7.20 has been changed to indicate the greater importance of internal stresses.

**Lines 7.8-11: I do not understand why there would be a warm anomaly if the southerly winds increase. I would expect the opposite, i.e. a cold anomaly as the air coming from the ice shelf is much colder.**

This counter-intuitive temperature anomaly linked to southerly winds is due to changes in low-level stratification. For example, during periods of low winds a layer of cool, dense air is observed near the surface, while during higher wind speeds the warmer overlying layer is mixed down toward the surface. Adiabatic warming linked to air traversing the TAM also contributes to this temperature anomaly. A similar description was added to the text at line 8.2-5.

**Lines 10.24-25: The 10 m wind speed in ERA-Interim should probably be given in the Methods section as well. More importantly here you state the AWS measures wind speed at a height of 2-3 m, which is inconsistent with lines 3.23, where you state that it measures wind speed at 10 m. Please correct either value and give a precise height.**

Corrected line 4.6, added ERA-Interim height to Methods section (5.18). Precise height for AWS is not possible due to snow accumulation on the RIS. The height of individual AWS will vary and is unable to be measured regularly.

**Line 13.11: SIC -> sea ice**

Change has been made at 14.11.

**Line 13.14: "reanalysis" -> "coarse resolution atmospheric reanalysis data"**

Suggested change has been made at 14.14

**Figure 1: Specify that grey lines show topography. Please add reference to source of topography. Also here the standard deviation should be calculated from the de-trended and de-seasonalized fields and only the period that is actually analyzed in the paper should be used for consistency. Please add that the red box is also the area used for Figure 2. Please add a legend for the wind rose.**

Figure 1 caption has been edited with reference added. If the April - October period is used here large variance is observed in northern regions due to summer sea ice break out. Even when de-seasonalised data is used increased variance is seen as the time breakout occurs varies by periods similar to the breakout period. For this reason the 20th Apil - 1st November period is used here.

**Figure 2: delete "dotted" for me these look like solid lines. "grey" -> "black"**

Figure has been edited.

**Figure 5 and 7: I guess these are composites. Please indicate this and the time period (years) over which the composite is formed.**

Period has been stated in text.

[revised manuscript text omitted]